# Electrophysiology of Endolysosomal Two-Pore Channels: A Current Account

**DOI:** 10.3390/cells11152368

**Published:** 2022-08-02

**Authors:** Sandip Patel, Yu Yuan, Cheng-Chang Chen, Dawid Jaślan, Gihan Gunaratne, Christian Grimm, Taufiq Rahman, Jonathan S. Marchant

**Affiliations:** 1Department of Cell and Developmental Biology, University College London, London WC1E 6BT, UK; y.yuan.12@ucl.ac.uk; 2Department of Clinical Laboratory Sciences and Medical Biotechnology, College of Medicine, National Taiwan University, Taipei 100229, Taiwan; chenchengchang@ntu.edu.tw; 3Department of Laboratory Medicine, National Taiwan University Hospital, Taipei 100225, Taiwan; 4Walther Straub Institute of Pharmacology and Toxicology, Faculty of Medicine, Ludwig-Maximilians University, 80336 Munich, Germany; dawid.jaslan@lrz.uni-muenchen.de (D.J.); christian.grimm@med.uni-muenchen.de (C.G.); 5Department of Cell Biology, Neurobiology and Anatomy, Medical College of Wisconsin, 8701 Watertown Plank Road, Milwaukee, WI 53226, USA; gunar005@d.umn.edu (G.G.); jmarchant@mcw.edu (J.S.M.); 6Department of Pharmacology, University of Cambridge, Cambridge CB2 1PD, UK; mtur2@cam.ac.uk

**Keywords:** TPC1, TPC2, TPCN1, TPCN2, NAADP, PI(3,5)P_2_, lysosomes, endosomes

## Abstract

Two-pore channels TPC1 and TPC2 are ubiquitously expressed pathophysiologically relevant proteins that reside on endolysosomal vesicles. Here, we review the electrophysiology of these channels. Direct macroscopic recordings of recombinant TPCs expressed in enlarged lysosomes in mammalian cells or vacuoles in plants and yeast demonstrate gating by the Ca^2+^-mobilizing messenger NAADP and/or the lipid PI(3,5)P_2_. TPC currents are regulated by H^+^, Ca^2+^, and Mg^2+^ (luminal and/or cytosolic), as well as protein kinases, and they are impacted by single-nucleotide polymorphisms linked to pigmentation. Bisbenzylisoquinoline alkaloids, flavonoids, and several approved drugs demonstrably block channel activity. Endogenous TPC currents have been recorded from a number of primary cell types and cell lines. Many of the properties of endolysosomal TPCs are recapitulated upon rerouting channels to the cell surface, allowing more facile recording through conventional electrophysiological means. Single-channel analyses have provided high-resolution insight into both monovalent and divalent permeability. The discovery of small-molecule activators of TPC2 that toggle the ion selectivity from a Ca^2+^-permeable (NAADP-like) state to a Na^+^-selective (PI(3,5)P_2_-like) state explains discrepancies in the literature relating to the permeability of TPCs. Identification of binding proteins that confer NAADP-sensitive currents confirm that indirect, remote gating likely underpins the inconsistent observations of channel activation by NAADP.

## 1. Introduction

Two-pore channels (TPCs) are ancient members of the voltage-gated ion channel superfamily that are localized to the endolysosomal system and related organelles in animals [1]. Structurally, they are modular proteins with two ‘shaker’-like ion channel domains arranged in tandem, each comprising six membrane-spanning regions. The first four (S1–S4) form a voltage-sensing domain. TPCs are dimers and, therefore, display pseudo-tetrameric symmetry, with S5–S6 from each domain contributing to the central pore (Figure 1). As such, they are likely descendants of a two-domain precursor which gave rise to four-domain ion channels such as voltage-gated Ca^2+^ channels upon intergenic duplication [2]. TPCs regulate numerous membrane trafficking events including but not limited to autophagy [3], transit of cargoes such as receptors [4] and viruses [5], pigmentation [6], neoangiogenesis [7], organelle morphology [8], membrane contact site formation [9], and phagocytosis [10] (as reviewed in [11,12]). They are increasingly implicated in diseases such as liver [4] and heart [13] dysfunction, Parkinson’s disease [8], and cancer [14] (as reviewed in [15,16]). Such links have prompted drug discovery efforts which are yielding a number of TPC modulators, including approved drugs [17,18]. 

TPCs in mammals came on the scene in 2009 when three independent studies all concluded that these proteins were the channels activated by NAADP [19,20,21]. NAADP is a Ca^2+^-mobilizing messenger long known to release Ca^2+^ not from the ER but rather from acidic organelles [22,23] in numerous cell types including neurons [24,25,26,27]. Therefore, the expression of TPCs in endolysosomes certainly fit the notion that they were NAADP targets, as did early electrophysiological evidence [28,29,30], as well as Ca^2+^ imaging experiments upon knockdown and overexpression of wildtype and mutant TPCs [19,20,21]. However, what was also clear early on was that NAADP did not bind directly to TPCs. Instead, results from photoaffinity labeling showed that NAADP bound small-molecular-weight proteins associated with TPCs [31,32]. Recent studies molecularly identified these long-sought receptors as JPT2/HN1L [33,34] and LSM12 [35], providing molecular evidence that NAADP does indeed function within a TPC complex [36].

However, a body of literature inconsistent with the NAADP–TPC–Ca^2+^ link accumulated since 2012, when TPCs were described as PI(3,5)P_2_-gated Na^+^ channels [37]. This evidence was based primarily on the elegant vacuolar patch-clamp technique. Under these conditions, TPCs were not activated by NAADP but rather by PI(3,5)P_2_. PI(3,5)P_2_ is a low-abundance phosphoinositide enriched in the endolysosomal system [38,39]. This lipid controls many aspects of membrane traffic, consistent with a role for TPCs in the same scenarios [38,39]. Notably, currents evoked by PI(3,5)P_2_ were Na^+^-selective and there was little evidence that TPCs were Ca^2+^-permeable. Structures of TPCs in complex with PI(3,5)P_2_ were resolved [40,41], providing snapshots of TPC opening, thus confirming that TPC2 is directly regulated by this lipid. Recent work has identified new small-molecule activators of TPC2 that mimic the disparate biophysical signatures of NAADP and PI(3,5)P_2_ [42,43]. Thus, TPCs are subject to polymodal gating and are highly unusual in functioning as both Ca^2+^-permeable channels in response to NAADP and Na^+^ channels in response to PI(3,5)P_2_ (Figure 1).

Here, we review the electrophysiological evidence underpinning TPC activation. We cover the techniques which have been applied to date (see also [44] in this issue) and summarize the insight gleaned from analysis of both macroscopic and microscopic (single-channel) currents. We also discuss how disparate datasets regarding the ion selectivity of TPCs and ligand activation have been reconciled through the discovery of TPC2 agonists and NAADP-binding proteins.

## 2. Methodology

Neher and Sakmann pioneered the ‘patch-clamp’ technique to measure ionic currents through channels in the plasma membrane of living cells (the Nobel Prize in Physiology or Medicine, 1991). It is considered the ‘gold standard’ for ion channel characterization. However, for intracellular ion channels such as TPCs, it is difficult to observe their electrophysiological profile directly using this approach. Nevertheless, TPCs have been characterized in their organellar locale using variations of the patch-clamp technique (Figure 2A), as well as by conventional recording upon rerouting TPCs to the plasma membrane (Figure 2B). The single-channel properties of TPCs have also been characterized in an organellar and cell surface setting, as well as in artificial or fused membranes (Figure 2C).

### 2.1. Macroscopic Recordings

A major advance in the recording of lysosomal ion channels was the development of the whole-endolysosome patch-clamp and the application of planar patch-clamp approaches [28,45]. Both require enlargement of the endolysosomal organelles up to 2–5 μm in diameter using chemical (e.g., vacuolin-1 and PIKfyve inhibitors) or molecular (e.g., Rab5-Q79L) tools [46]. This increase in size then allows measurement in a manner similar to conventional electrophysiological recordings (Figure 2A). The details of the methodology have previously been described [47,48]. In brief, vacuolin-1 pretreatment is the most commonly used method to enlarge the endolysosome. In vacuolar patch-clamp analyses, a single enlarged vesicle is physically pushed out of the cell by creating a breach in the cell surface using a micropipette to allow access by the recording pipette. In planar patch-clamp analyses, the vesicles are isolated biochemically and recorded on a glass chip containing a microstructured hole. In both cases, pressure/suction is used to form tight (GΩ) seals between the organelle membrane and the edge of the glass aperture. The resulting ‘whole-organelle’ currents are equivalent to whole cell currents in conventional recording, but where the pipette interior or internal recording chamber corresponds to the lysosome lumen. Whole-organelle currents represent the total channel activity across the entire membrane surface. TPCs have also been recorded upon expression in plant and yeast vacuoles, which are large enough to patch directly [49].

TPCs are targeted to the endolysosomal system via dileucine-based motifs [29]. Mutation or deletion of this sequence in TPC2 results in expression of TPC2 in the plasma membrane. This allows the use of the conventional patch-clamp approach for recording TPC2 activity [29]. TPCs have been characterized both in the whole-cell mode where the pipette solution corresponds to the cytosol and in the inside-out mode where the pipette solution corresponds to the lysosome lumen (Figure 2B).

### 2.2. Microscopic (Single-Channel) Recordings

Single-channel recordings of TPCs have been achieved using several techniques (Figure 2C). This includes those for cell-surface TPC2 in the inside-out mode as described above [29]. Here, one or sometimes more TPCs in a membrane patch are captured within the tip of the capillary glass that constitutes the patch pipette. Microscopic TPCs currents have also been recorded using the commonly used bilayer technique [50]. In this method, immunoprecipitated TPCs [30] or endolysosomal vesicle preparations expressing TPCs [51] are embedded in an artificial lipid membrane usually comprising a mixture of synthetic 1-palmitoyl-2-oleoyl-glycero-3-phosphocoline (POPC) and 1-palmitoyl-2-oleoyl-glycero-3-phosphoethanolamine (POPE). This membrane is formed across an aperture in a partition that forms two open chambers.

TPC activity at the single-channel level has also been captured in nuclear and lysosome membranes. Curiously, stably expressed TPC2 in IP_3_ and ryanodine-receptor-deficient chicken B lymphocytes localized to the ER and the nuclear membrane instead of endolysosomes [52]. However, this mistargeting allowed recording of currents from isolated intact nuclei in the ‘on nucleus’ mode similar to the cell-attached mode in conventional patch-clamping. TPC-like currents have also been recorded from giant unilamellar vesicles formed from isolated lysosomal fractions [53].

In sum, several independent electrophysiological approaches, some novel, have been applied to characterize the channel properties of TPCs.

## 3. Lysosomal Patch-Clamp Analyses of TPCs

The first electrophysiological recordings of TPCs were reported by Schieder et al. [28]. for TPC2 using the planar patch-clamp technique In this study, NAADP induced Ca^2+^ currents in enlarged lysosomes expressing mouse TPC2 but not a pore mutant (N257). These data established the sensitivity of the channel to NAADP in an organellar setting. Subsequent studies reported the first whole-lysosome recordings of TPCs [37,54]. This work revealed that, like TRPML1 [55], both TPC1 and TPC2 were activated by PI(3,5)P_2_ but not NAADP. Again, mutation of a conserved pore residue (D276; human TPC2) abrogated channel activity [37]. A subsequent study by Grimm et al. [4] confirmed the sensitivity of murine TPC2 to both NAADP and PI(3,5)P_2_ using planar patch-clamp analyses. Unlike TPC2, TPC1 is a voltage-sensitive channel dependent on arginine residues in the putative voltage sensors [56].

### 3.1. Regulation of TPCs

NAADP-evoked Ca^2+^ currents through TPC2 in planar patch-clamp analyses were only observable at acidic luminal pH [28]. In contrast, PI(3,5)P_2_-evoked Na^+^ currents in vacuolar patch-clamp analyses were pH-insensitive [37]. Vacuolar patch-clamp analyses revealed that TPC1 and TPC2 were inhibited by ATP—an effect ascribed to regulation of the channels by mTOR [54]. mTOR immunoprecipitated with both TPCs, and this was subsequently confirmed by nonbiased proteomics [6]. However, the molecular basis for this regulation remains to be established, as no phosphorylation site was identified. Inhibition of PI(3,5)P_2_-mediated currents by ATP was confirmed by Jha et al. [57] for TPC2 using the same approach. However, they revealed additional negative regulation of TPC2 by JNK kinases. Again, the molecular basis for this regulation was not defined. Jha et al. [57] also demonstrated marked inhibitory effects of Mg^2+^ on TPC2 activation by both PI(3,5)P_2_ and NAADP. Cytoplasmic Mg^2+^ selectively inhibited outward (lysosomal-directed) currents. In contrast, luminal Mg^2+^ inhibited both inward and outward currents but only at neutral (non-physiological) luminal pH. Increases in luminal pH stimulated voltage activation of TPC1 [56] (see [51]; Section 5).

The above studies were based on electrophysiological analysis of TPCs expressed in enlarged lysosomes in mammalian cells. However, TPCs have also been expressed heterologously in plants and yeast where they are targeted to vacuoles [49]. Analyses of both human TPC isoforms have been performed in *Arabidopsis* vacuoles. *Arabidopsis* possesses a TPC gene that is insensitive to PI(3,5)P_2_ and NAADP [49]. In contrast, human TPC2 expressed in plants targeted to the vacuole and retained its sensitivity to PI(3,5)P_2_ but not NAADP [49]. Leveraging the differential sensitivity of plant and animal TPCs to PI(3,5)P_2_, Kirsch et al. [58] performed molecular dynamics simulations to predict the PI(3,5)P_2_-binding site in human TPC2. This analysis identified a pocket formed by the S2–S3 and S4–S5 linkers in domain I and the S6 regions in both domains. Functional analyses of mutant TPC2 channels in which predicted binding site residues were in general substituted for the plant counterparts were performed in plant vacuoles devoid of endogenous TPC. This elegant analysis confirmed a key requirement for lysine residues in the S4–S5 linker (K203, K204, and K207) and a serine residue in S6 (S322) of domain I. Introduction of negative charges in neighboring residues predicted not to interact with PI(3,5)P_2_ also reduced activity, which the authors ascribed to electrostatic repulsion of the incoming lipid [58].

Analysis of human TPC1 in TPC knockout *Arabidopsis* vacuoles confirmed PI(3,5)P_2_ and voltage sensitivity and revealed regulation of the channel by Ca^2+^ [59]. Cytosolic Ca^2+^ activated the channel in the low micromolar range by shifting the voltage dependence to more hyperpolarized potentials. Conversely, luminal Ca^2+^ (10, 1000 µM) had the opposite effect. This inhibition is reminiscent of that for plant TPC where Ca^2+^ inhibits channel activity through a structurally and functionally resolved inhibitory binding site which accounts for the gain-of-function *fou2* mutation [60,61,62,63,64]. PI(3,5)P_2_ activation of human TPC2 was also demonstrated upon channel expression in giant vacuoles from yeast [49].

### 3.2. Pharmacology of TPCs

Biophysical analyses of organellar TPCs have provided insight into their pharmacology. Both TPC2 [37,49] and TPC1 [56] are blocked by verapamil in vacuolar patch-clamp analyses of enlarged lysosomes and/or plant vacuoles. Verapamil is a blocker of voltage-gated Ca^2+^ channels but also NAADP action [65], most likely reflecting shared ancestry between voltage-gated Ca^2+^ channels and TPCs [2]. Curiously, PI(3,5)P_2_-evoked currents through both TPCs are also blocked by the NAADP antagonist Ned-19 albeit at high concentrations (100–200 µM) [5,37]. The introduction of tetrandrine as a blocker of TPCs based on its potent anti-EBOV activity was demonstrated for both NAADP and PI(3,5)P_2_ activation of TPCs using planar and vacuolar patch-clamp analysis, respectively [5]. Screening of tetrandrine analogues identified SG-094 as a more potent derivative in cell proliferation assays with anticancer activity in vivo and blocking activity against PI(3,5)P_2_-evoked currents in TPC2-expessing vacuoles [66]. Expression of both human TPC isoforms in plant vacuoles also demonstrated (low-affinity) block by the flavonoid naringenin and possible voltage-dependent regulation by Mg^2+^ [67], as reported by Jha et al. [57]. A screening campaign of other flavonoids, again using cell proliferation as a primary readout, subsequently confirmed them as TPC2 blockers using vacuolar patch-clamp analysis of TPC2 activated by PI(3,5)P_2_, with MT-8 being the most potent [68]. Results from a structure-based virtual screen against the pore region of TPC2, cross-referenced against physical screens of approved drugs that blocked Ebola virus entry, identified a number of selective estrogen receptor modulators and D2 receptor antagonists as TPC2 blockers [17]. Representative drugs from these classes (raloxifene and fluphenazine) inhibited TPC2 currents evoked by PI(3,5)P_2_ in enlarged lysosomes [17]. In the hunt for TPC2 activators, Zhang et al. [18] performed a high-throughput Ca^2+^-based screen and identified a number of approved drugs that evoked Ca^2+^ signals in cells stably expressing TPC2. Subsequent vacuolar measurements confirmed agonistic activity albeit with low affinity on TPC2 [18]. This analysis singled out two groups of tricyclic antidepressants (TCAs) and riluzole as TPC2 agonists. Strikingly, whereas currents evoked by riluzole were linear and voltage-insensitive, similar to PI(3,5)P_2_, TCA-induced currents exhibited strong inward rectification, thereby revealing voltage dependence. Moreover, clomipramine synergized with PI(3,5)P_2_ to activate TPC2, whereas the effects of riluzole and PI(3,5)P_2_ were additive. These drugs had complex effects on TPC1, with clomipramine activating the channel by shifting the voltage dependence to more hyperpolarizing potentials, and with riluzole and a structurally distinct TCA, chlorpromazine, causing inhibition.

### 3.3. TPC Variants

Genome-wide association studies prior to the emergence of TPCs as NAADP targets identified two single-nucleotide polymorphisms in TPC2 (M484L and G734E) associated with blonde hair [69]. Subsequent electrophysiological analysis indicated that both were gain-of-function variants [70]. The M484L variant displayed a decrease in the EC_50_ for PI(3,5)P_2_ activation and an increase in the maximal current amplitude. Synthetic TPC2 agonists (see Section 7) were also more effective in cells expressing the M484L variant [42]. Molecular dynamics simulations suggested that pore dilation likely accounted for ion selectivity changes to monovalent ions [70]. Interestingly, activation was only apparent in the background of another, more common SNP (L564P) [71], suggesting that L484 and P564 functionally interact in some way. In contrast, PI(3,5)P_2_-evoked currents for G734E variant were not different to the wildtype channel [70]. However, the variant was less sensitive to ATP inhibition and activated to a greater extent by mTOR inhibitors [70], suggesting functional relevance during starvation.

### 3.4. Endogenous TPC Activity

Importantly, organellar recordings have confirmed an action of NAADP and PI(3,5)P_2_ on endogenous TPCs in a number of cell types (Table 1). These include macrophages (both peritoneal and bone-marrow-derived), cardiomyocytes, embryonic fibroblasts, and brain microglia from mice [4,37,43,54,56,72]. In these studies, responses to TPC activators were compared in enlarged lysosomes from wildtype and TPC knockout animals. Endogenous TPC2-like currents in response to PI(3,5)P_2_ have also been characterized in adult fibroblasts confirming gain-of-function variants [70,71], as well as in human melanocyte [73] and mouse liver cancer [66] cell lines in which TPC2 was targeted using CRISPR/Cas9. In the former case, the currents were recorded from melanosomes. Endogenous TPC2-like PI(3,5)P_2_-evoked currents that were tetrandrine-sensitive were also characterized in T24 cells derived from a human bladder cancer [14]. Currents induced by clomipramine (a TCA) and riluzole were abolished in HAP1 cells knocked out for both TPCs [18].

In sum, organellar recordings have defined the channel properties of both recombinant and endogenous TPCs in a near-native setting and provided key insight into their regulation by endogenous and pharmacological cues.

## 4. Conventional Patch-Clamp Analyses of TPCs

Brailoiu et al. [29] performed the first conventional patch-clamp recordings of TPCs by identifying an N-terminal dileucine lysosomal-targeting motif, mutating it in TPC2 (L11A/L12A) and recording in the whole-cell mode from mutant channels rerouted to the plasma membrane. This study demonstrated that TPC2 was an NAADP-gated Ca^2+^-permeable channel. Subsequent studies by Wang [37] and Jha [57] et al. used the same approach to measure TPC2 activity. In the whole-cell and inside-out macro-patch recordings of Wang et al. [37], TPC2 was insensitive to NAADP and instead sensitive to PI(3,5)P_2_ and Na^+^-selective. This profile was consistent with whole-endolysosome recordings (Section 3). In contrast, Jha et al. [57] reported sensitivity of TPC2 to both NAADP and PI(3,5)P_2_ and reported similar inhibition by Mg^2+^ and P38/JNK in plasma membrane patches to parallel vacuolar recordings [57]. Currents were readily detectable at neutral luminal pH [29] and pH-insensitive in the absence of Mg^2+^ [57].

Guo et al. [75] made parallel recordings of cell-surface human TPC2 (inside out) and plant TPC1 (whole cell) expressed in HEK cells TPC2 was again shown to be a PI(3,5)P_2_- but not NAADP-gated, highly Na^+^-selective channel similar to the studies of Wang et al. [37]. In contrast, plant TPC1 behaved as a voltage-gated nonselective cation channel permeable to Ca^2+^ under the specified conditions. In the face of such ion selectivity differences, Guo and colleagues ‘swapped’ divergent residues within the selectivity filters of the two channels. In an elegant set of results, the Na^+^ selectivity of TPC2 relative to Ca^2+^ and K^+^ was reduced, whereas that for plant TPC was increased. Structural analyses of ‘humanized’ plant TPC1 rationalized Na^+^ selectivity at the atomic level.

She and colleagues resolved the structures of mammalian TPCs in complex with PI(3,5)P_2_ using cryo-electron microscopy and complemented their studies with functional measurements of TPCs at the plasma membrane in whole-cell or inside-out configurations [40,41]. Like human TPC2, mutation of the N-terminal targeting sequence in mouse TPC1 revealed currents at the plasma membrane [40]. Notably, deletion of the sequence in human TPC1 did not appear to affect channel localization [29]. Nevertheless, in their recordings, both TPC isoforms formed PI(3,5)P_2_-sensitive Na^+^-selective channels upon re-routing [40,41]. Mutagenesis showed that TPC activation by PI(3,5)P_2_ required a cluster of arginines (TPC1) or lysines (TPC2) in the S4–S5 linker and select S6 residues in domain I [40,41], consistent with the work of Kirsch et al. [58]. Mechanistically, the linker served as the main binding pocket for PI(3,5)P_2_, while S6 residues transmitted conformational changes leading to pore opening. TPC1 was additionally voltage-sensitive, requiring Arg540 and Arg546 residues in S4 of domain II [40,76]. In contrast, TPC2 was voltage-insensitive due to absence of one of these critical residues [41]. Indeed, introducing an arginine at the equivalent position in TPC2 conferred voltage sensitivity [41]. Voltage-dependent and -independent activation of TPC2 by TCAs and riluzole in vacuoles was confirmed in whole-cell patch-clamp analyses of TPC2 expressed at the cell surface.

Overall, rerouting TPCs to the plasma membrane has provided a convenient means for patch-clamp recording.

## 5. Single-Channel Analyses of TPCs

Compared to macroscopic recordings of TPCs, there have been a limited number of studies investigating the microscopic properties of TPCs at the single-channel level, and most to date have characterized NAADP-mediated channel activity.

The first single-channel recordings of TPCs focused on TPC2 reconstituted in artificial lipid bilayers [30] and in membrane patches excised from the plasma membrane of HEK-293 cells expressing rerouted TPC2 [29]. These early studies established permeability of TPC2 to Ca^2+^ albeit with variable single-channel conductances (Table 2), as well as to monovalent cations including K^+^ and Cs^+^ [29,30] upon NAADP stimulation. Importantly, mutation of a conserved leucine residue within the pore (L265P) substantially reduced Cs^+^ conductance, providing early evidence that TPC2 is indeed the pore-forming subunit mediating NAADP action [29], consistent with mutagenesis and the macroscopic recordings of Schieder et al. [28]. In accordance, manipulating TRPML1 activity (a proposed NAADP target) had little effect on TPC2 channel activity [77]. Conductance of TPC2 to monovalents was confirmed for NAADP-stimulated TPC2 by nuclear patch-clamping [52] and more recently extended to Na^+^ in plasma membrane patches [17] (Table 2).

Bilayer studies with NAADP-stimulated TPC1 also established the permeability of TPC1 to Ba^2+^ [51], which is often used as a Ca^2+^ surrogate, as well as Ca^2+^ itself [78], and to monovalents (K^+^ and Na^+^), including interestingly H^+^ (although the single-channel conductance was not reported) [78]. The single-channel Na^+^ conductance of TPC1 expressed in plant vacuoles upon voltage activation has also been estimated by nonstationary noise analysis [79]. Again, mutation of the pore with the equivalent leucine in TPC1 (L273) abrogated activity evoked by NAADP [51]. Of note, using Ba^2+^ as the charge carrier, two conducting states with markedly differing apparent γBa^2+^ were observed for TPC1 in bilayers [51] (Table 2). The two states were kinetically tightly coupled and appeared to represent intrinsically distinct conductance states as opposed to concerted opening of several TPC1 channels [51]. Multiple conductance states for TPC2 have also been reported [30].

A hallmark feature of NAADP action in mammalian cells is its bell-shaped concentration–effect relationship [80]. Thus, whereas low (nM) concentrations of NAADP evoke concentration-dependent increases in activity, higher (micromolar) concentrations evoke concentration-dependent decreases [81]. This behavior has been recapitulated at the single-channel level for TPC2 in bilayers [30] and in nuclear patches [52], where open probability was biphasically regulated by NAADP. Kinetically, regulation of TPC2 open probability by NAADP was accounted for by biphasic effects on the channel’s mean open time [52]. However, in excised plasma membrane patches expressing TPC2, NAADP-mediated increases in TPC2 open probability were independent of changes in mean open time [17]. The exact reason for such discrepancy remains unclear, but it is worth mentioning that the two studies involved different membrane environments (nucleus vs. plasma membrane) and permeating ions (K^+^ vs. Na^+^).

Single-channel activity of TPC2 in bilayers is positively regulated by luminal Ca^2+^ (10 µM–1 mM) [30]. So too is TPC1 activity (EC_50_ 180 µM) [51], although analyses of PI(3,5)P_2_-stimulated macrocurrents upon expression of TPC1 in plant vacuoles revealed inhibition by luminal Ca^2+^ [59]. TPC1 also appears to be gated by cytosolic Ca^2+^ with no further increases in channel activity upon NAADP stimulation [78]. Positive regulation of single-channel activity by Ca^2+^ is reminiscent of macroscopic recordings of PI(3,5P)_2_-mediated channel activity [59]. Intriguingly, the sensitivity of TPC1 to NAADP appears to also be regulated by voltage such that the EC_50_ for channel activation by NAADP was decreased at hyperpolarizing potentials [51]. Put another way, activation of the channel by depolarization would be inhibited in the presence of NAADP. This contrasts to depolarization-induced activation of TPC1, which is stimulated by PI(3,5)P_2_ [56], raising the possibility that TPC1 may be reciprocally regulated by its ligands through interplay with voltage.

Like voltage, pH also has complex effects on TPCs. On the one hand, NAADP-gated single-channel activity of TPC1 in bilayers is enhanced at acidic pH [51]. On the other hand, macroscopic current recordings of voltage-gated TPC1 activity are negatively regulated by acidic pH [56]. For TPC2, the concentration–effect relationship for NAADP is, as stated, biphasic but only at acidic pH [30]. Overall, however, Po was decreased at acidic pH consistent with the planar patch-clamp analyses [28] but inconsistent with vacuolar [37] and plasma membrane patch-clamp analyses [57].

TPC2 is also reportedly regulated by cyclic-AMP-dependent protein kinase [52] but the proposed phospho-site (S666) is found within the pore and, thus, likely inaccessible to kinases. The NAADP antagonist Ned-19 blocked single TPC2 channels [29,30]; however, in the studies of Pitt et al. [30], it activated TPC2 at lower, nanomolar concentrations. The TPC blockers raloxifene and fluphenazine inhibited single-channel activity of TPC2 by reducing the open time consistent with an action within the pore [17]. In contrast, the NAADP antagonist Ned-19, which similarly reduced single-channel activity of TPC2, did so independently of changes in open time [17]. This is consistent with a more remote action of Ned-19 (and NAADP) on the channel, most likely at the level of the receptor (see Section 6).

All-important endogenous NAADP-mediated single-channel activity has been recorded from endolysosomal vesicles prepared from HEK cells and incorporated into bilayers [51] (Table 1). Here, NAADP evoked both small and large conductance activity using Ba^2+^ as the permeant ion similar to overexpressed TPC1. Activity was sparse but reduced by both siRNA-mediated knockdown of TPC1 and overexpression of TPC1 L273P, consistent with it being mediated by TPC1 [51]. Endogenous NAADP-mediated activity has also been recorded from lysosome vesicles, again isolated from HEK cells but incorporated into giant unilamellar vesicles [53]. This activity was attributed to TPC2, but TPC2 manipulations to confirm this were lacking. Indeed, NAADP activity was characterized after chelation of cytosolic Ca^2+^ because of considerable spontaneous activity. Such Ca^2+^ dependence is reminiscent of TPC1 [78].

In sum, single-channel recordings, albeit limited, have defined the fundamental properties of both recombinant and endogenous TPCs at the highest resolution possible.

## 6. Conflict Resolution: On the Identification of TPC2 Agonists and NAADP-Binding Proteins

Ion channels are defined by their ion selectivity and activators. However, the biophysical data on TPCs are divided more or less into two camps: the original ‘calcium’ camp advocating TPCs as Ca^2+^-permeable and NAADP-gated, and the rival ‘sodium’ camp advocating TPCs as Na^+^-selective and PI(3,5)P_2_-gated. This is summarized in Table 3, which collates the relative permeabilities of TPCs to Ca^2+^, Na^+^, and K^+^ and their sensitivity to NAADP and PI(3,5)P_2_. Because TPCs were originally viewed in the context of NAADP-mediated Ca^2+^ signaling, the initial electrophysiological focus was naturally on Ca^2+^ permeability. However, channel activation by PI(3,5)P_2_ switched focus to its Na^+^ permeability. Overall, there was congruence on the relative permeability within the camps (although data on NAADP were limited) but not between them. What is the reason for the differences in ion selectivity and ligand sensitivity between camps?

It was the discovery of two structurally distinct small-molecule activators of TPC2 (TPC2-A1-N and TPC2-A1-P) which provided a likely explanation [42]. These activators were discovered in a high-throughput (Ca^2+^-based) screen, confirming the Ca^2+^ permeability of TPC2, thus being consistent with the calcium camp. TPC2-A1-N evoked rapid influx in cells expressing cell-surface TPC2 and robust Ca^2+^ release in cells expressing lysosomal TPC2-GCaMP6s. Electrophysiological analyses using vacuolar patch-clamp revealed that TPC2-A1-N currents were essentially nonselective. In contrast, TPC2-A1-P evoked sluggish Ca^2+^ influx responses and was largely ineffective in lysosomal Ca^2+^ release assays. Moreover, electrophysiological analyses of TPC2-A1-P action showed that it evoked larger currents relative to TPC2-A1-N that were Na^+^-selective. These markedly different electrophysiological profiles are highly reminiscent of those reported for NAADP (when observable) and PI(3,5)P_2_, respectively (Table 3). Indeed, parallel measurements of NAADP- and PI(3,5)P_2_-evoked currents confirmed the Ca^2+^ and Na^+^ modalities. What emerged was that the ion selectivity of TPC2 is agonist-dependent. This challenges conventical wisdom where the ion selectivity of a given ion channel is thought to be fixed, but readily explains apparently contradictory findings in the literature relating activation of TPCs by NAADP and PI(3,5)P_2_. They, thus, unite the calcium and sodium camps. The effects of TPC2-A1-P but not TPC2-A1-N on TPC2 were reduced by mutation of K204 [42], which is essential for PI(3,5)P_2_ activation [41]. Notably, currents evoked by the TPC2 activators identified by Zhang et al. [18] were all Na^+^-selective; thus, PI(3,5)P_2_-like but channel activation was independent of K204, suggesting a distinct binding pocket to TPC2-A1-P and PI(3,5)P_2_.

The above findings indicating that TPC2 behaves as a nonselective cation channel activated by NAADP but as a Na^+^-selective channel activated by PI(3,5)P_2_ are readily apparent when looking back at the literature (Table 3). The problem with accepting this (aside from the challenge of dogma) was that there were few studies until recently that directly compared ion selectivity to NAADP and PI(3,5)P_2_ in parallel. This was almost exclusively related to the findings by some labs that TPC2 was NAADP-insensitive, thereby excluding side-by-side analyses with PI(3,5)P_2_. In this context, studies by Pitt et al. [78] noted a modest shift in the permeability ratio when TPC1 in bilayers was stimulated with NAADP in the presence of PI(3,5)P_2_, although the channel was not demonstrably gated by PI(3,5)P_2_. Ogunbayo et al. [82] also noted differences in the properties of vacuolar TPC2 currents by NAADP and PI(3,5)P_2_ in Na^+^- and Ca^2+^-rich recording solutions, although the NAADP-mediated currents were small.

Thus, why is NAADP sensitivity patchy? The discovery of NAADP-binding proteins that confer NAADP sensitivity provides a potential explanation. Using a next-generation photoaffinity NAADP probe, both Gunaratne et al. [33] and Roggenkamp et al. [34] converged on JPT2 (also known an as HN1L) as an NAADP-binding protein from protein purification of red blood cells and T lymphocytes, respectively. Work by Zhang et al. [35] used a proteomic approach to identify LSM12 as a distinct NAADP-binding protein. Both are predicted soluble proteins, both were necessary for NAADP-mediated Ca^2+^ signaling, and both coimmunoprecipitated with TPCs as anticipated from early photoaffinity labeling studies [83]. Importantly, NAADP-evoked currents through cell-surface TPC2 were abolished upon LSM12 knockout and restored upon LSM12 re-expression or co-injection of recombinant LSM12 with NAADP [35]. Similar conclusions were obtained with TPC1 rerouted to the plasma membrane, although the properties of the current were not reported.

The separation of receptor from channel could result in differential loss/absence of the former in functional NAADP assays. Indeed, in contrast to the studies of Brailoiu et al. [29] and Jha et al. [77], there were no detectable NAADP-evoked currents in excised plasma membrane patches expressing TPC2 in the recent work by Zhang et al. [35]. Rather, NAADP sensitivity was only revealed upon combining whole-cell recording with NAADP microinjection, highlighting the fickle nature of NAADP responsiveness. This problem might be more acute in broken cell preparations such as vacuolar patch-clamps that would dilute the binding proteins and promote dissociation from any preformed complexes, thereby negating NAADP action. However, NAADP sensitivity in bilayers is readily demonstrable despite extensive biochemical manipulation prior to incorporation.

In sum, recent advances in the identification of TPC2 agonists and NAADP-binding proteins go a long way in reconciling reported differences in ion selectivity and ligand sensitivity of TPCs.

## 7. Outlook

Despite their inaccessibility relative to cell-surface channels, TPCs have been interrogated electrophysiologically, providing direct insight into channel function. Such insight has been focused on TPC2. Thus, more work is needed for TPC1 and establishing how ligand activation by NAADP and PI(3,5)P_2_ is integrated with voltage activation. In this context, TPC3, which completes the ancestral TPC family, is also voltage- and likely NAADP- and phosphoinositide-sensitive [84,85,86], but almost completely ignored [87], probably because of its absence in mice and men [84,88]. Endogenous TPC activity has been characterized in numerous cells as discussed (Table 1) yet mostly limited to PI(3,5)P_2_. Thus, more studies are required for NAADP.

Vacuolar patch-clamp analyses of TPCs are technically demanding and low-throughput. TPCs rerouted to the plasma membrane offer a more facile route to channel activation. Albeit in an alien environment, cell-surface TPCs recapitulate many aspects of the channel in its lysosomal environment, as discussed. However, it should be stressed that the lysosomal environment could also be considered alien given the necessary enlargement required. Planar patch-clamp analysis circumvents manual manipulations but has not been taken up by the community to the extent of other techniques. In this context, imaging approaches using, e.g., TPC2-GCaMP6s fusions [42] do not require vacuolar enlargement.

Single-channel analyses have been forthcoming in providing details into the fundamental channel behavior. To date, such analyses have focused almost exclusively on the actions of NAADP. Therefore, information on PI(3,5)P_2_ is required not least due to the very different effects of the ligands on ion selectivity. One curiosity that needs explaining is the permeability of both TPC isoforms to K^+^ in microscopic (Table 2) but not macroscopic (Table 3) recordings. The reported effects of luminal pH on TPCs also vary wildly.

The identification of cell-permeable NAADP and PI(3,5)P_2_ mimetics that switch ion selectivity and NAADP-binding proteins that associate with TPCs represent step changes in our understanding of how TPCs are activated. Key now is to define the relationship between direct and indirect channel activation by these activators at the channel level. Indeed, electrophysiological characterization of TPC in complex with JPT2 is currently lacking. Advances in the chemical and molecular biology of TPCs will certainly facilitate this.

## Figures and Tables

**Figure 1 cells-11-02368-f001:**
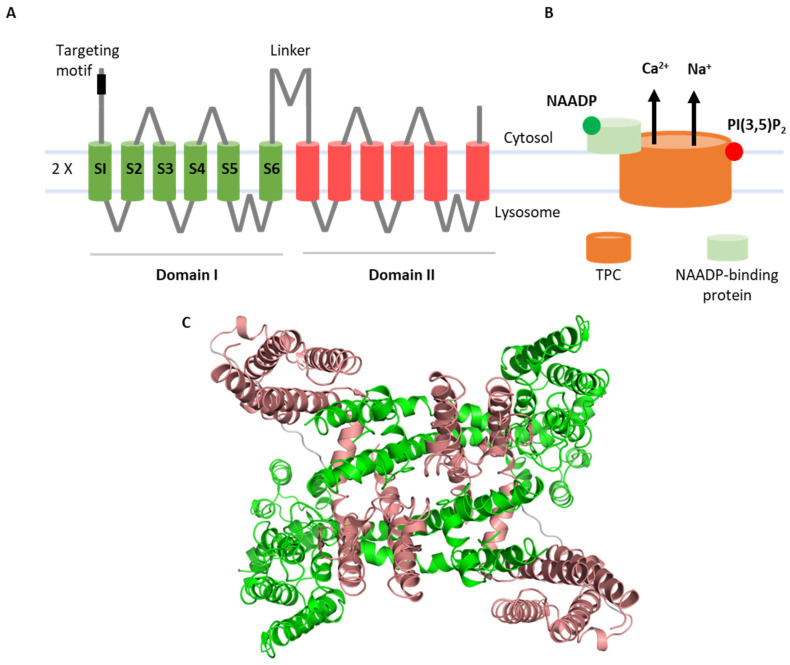
Structure of TPCs. Cartoon illustrations showing domain organization (**A**) and activation scheme (**B**) of TPCs. Cryo-EM structure of TPC2 (PDB: 6nq0) where individual channel domains are color-coded (**C**).

**Figure 2 cells-11-02368-f002:**
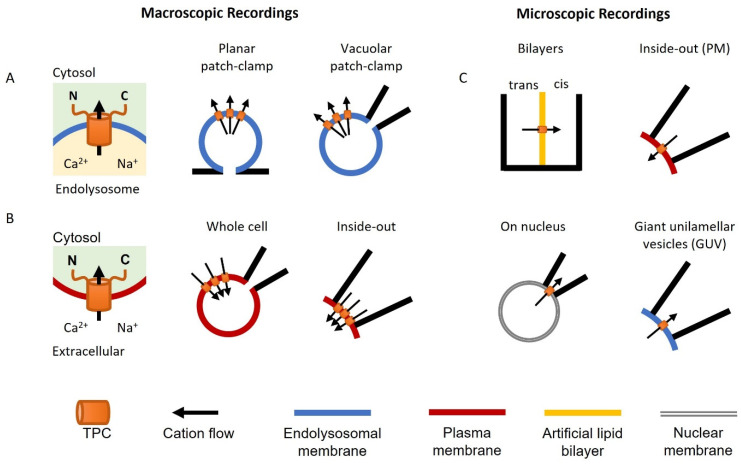
Electrophysiological methods used to characterize TPCs. Cartoon illustrations showing planar and vacuolar patch-clamp analyses of endolysosomal TPCs (**A**), conventional patch-clamp of plasma membrane TPCs in whole-cell and inside-out configurations (**B**), and single-channel analyses of TPCs imbedded in bilayers, isolated nuclei, the plasma membrane (PM), and giant unilamellar vesicles (GUV) (**C**).

**Table 1 cells-11-02368-t001:** Endogenous TPC channel activity. ^1^ Melan-a cell line derived from Oa1^−/−^ mice. Hsa, *Homo sapiens*; Mmu, *Mus musculus*; Lca, *Lithobates catesbeianus*.

Isoform	Cell Type	Activator	Validation Method	Reference
HsaTPC1	HEK cells	NAADP	siRNA, dominant negative	[51]
MmuTPC2	Macrophages	PI(3,5)P_2_	Double knockout mouse	[37]
MmuTPC1	Macrophages	PI(3,5)P_2_	Knockout mouse	[54]
MmuTPC2	Macrophages	PI(3,5)P_2_	Knockout mouse	[54]
MmuTPC1/2	Cardiomyocytes	PI(3,5)P_2_	ATP block	[54]
	Hepatocytes			
	Fibroblasts			
MmuTPC1	Cardiomyocytes	PI(3,5)P_2_	Knockout mouse	[56]
MmuTPC2	MEFs	PI(3,5)P_2_	Knockout mouse	[4]
MmuTPC2	MEFs	NAADP	Knockout mouse	[72]
MmuTPC2	Melanocytes ^1^	PI(3,5)P_2_	CRISPR/Cas9 knockout	[73]
LcaTPC2	RPE cells	PI(3,5)P_2_		[73]
HsaTPC2	HEK cells	NAADP		[53]
HsaTPC2	Adult fibroblasts	PI(3,5)P_2_	WT vs. M484L	[70]
HsaTPC2	T24 cells	PI(3,5)P_2_	Tetrandrine block	[14]
HsaTPC1/2	HAP1 cells	Clomipramine	CRISPR/Cas9 knockout	[18]
MmuTPC2	Macrophages	TPC2-A1-N/P	Knockout mouse	[43]
MmuTPC2	RIL175 cells	PI(3,5)P_2_	CRISPR/CAS9 knockout	[66]
MmuTPC2	Macrophages	PI(3,5)P_2_	Knockout mouse	[74]
	Microglia			
HsaTPC2	Adult fibroblasts	PI(3,5)P_2_	WT vs. K376R/G387D	[71]

**Table 2 cells-11-02368-t002:** Single-channel conductance of TPCs. Data are reported for NAADP or voltage ^1^ activation, along with ^2^ endogenous activity.

Isoform	Ca^2+^	Ba^2+^	Na^+^	K^+^	Cs^+^	Reference
HsaTPC1		50/200 pS				[51]
HsaTPC1	19 pS		68 pS	87 pS		[78]
HsaTPC1 ^1^			5.5 pS			[79]
HsaTPC2	15 pS			300 pS		[30]
HsaTPC2	40 pS				128 pS	[29]
HsaTPC2					100 pS	[77]
HsaTPC2				208 pS	78 pS	[52]
HsaTPC2 ^2^				207 pS		[53]
HsaTPC2			85 pS			[17]

**Table 3 cells-11-02368-t003:** Agonist-dependent ion selectivity in TPCs. Reported cation permeability ratios for endolysosomal TPC1 and TPC2 activated by NAADP and PI(3,5)P_2_. Mean *P*_Ca_/*P*_Na_ values for TPC2 are compared to recent work [43], where the actions of NAADP and PI(3,5)P_2_ were assayed in parallel. ^1^ Ba^2+^ used instead of Ca^2+^ as the permeant ion. ^2^ Endogenous activity. ^3^ Values were calculated on the basis of an estimated reversal potential of −82 mV (intracellular [Na^+^] = 0.2 M, extracellular [Ca^2+^] = 0.05 M) using the bi-ionic equation in [42] and −80 mV (intracellular [Na^+^] = 0.2 M, extracellular [K^+^] = 0.1 M) using the bi-ionic equation described in [75]. ^4^ Recorded from melanosomes.

Isoform	Sensitive?	NAADP *P*_Ca_/*P*_Na_	*P*_Ca_/*P*_K_	Sensitive?	PI(3,5)P_2_ *P*_Na_/*P*_Ca_	*P*_Na_/*P*_K_	Reference
HsaTPC1	√		2.2 ^1^				[51]
HsaTPC1	√	0.98	0.11				[78]
MmuTPC1	×			√	212.3	78.1	[56]
HsaTPC1	×			√	10–20	35.7	[59]
MmuTPC1				√	11.4	65.8	[40]
MmuTPC2	√		>1000				[28]
HsaTPC2	√		2.6				[30]
MmuTPC2 ^2^	√	0.7	340	√			[4]
MmuTPC2 ^2^	√	0.57, 0.86	286				[72]
HsaTPC2	×			√	10	33.3	[37]
HsaTPC2 ^3^	×			√	~12	~11	[49]
MmuTPC2 ^4^				√	≥100	≥50	[73]
HsaTPC2	×			√	16.8	23.8	[75]
HsaTPC2				√		~10	[70]
	*P*_Ca_/*P*_Na_ = 0.76 ± 0.1			*P*_Ca_/*P*_Na_ = 0.06 ± 0.02			
	III			III			
HsaTPC2	√	0.73		√	0.08		[43]

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
