# Peer review of "Electrophysiology of Endolysosomal Two-Pore Channels: A Current Account"

_cells, 2022, doi:10.3390/cells11152368_

Round 1

Reviewer 1 Report

In this review, Patel at al. focused on the electrophysiological recordings of two-pore channels TPC1 and TPC2, highlighting known and novel intriguing experimental results. The paper is clear, easy to follow and very informative.

There are some references, which should be included, and some minor changes to be made.

- Festa et al. (Cells. 2022 Mar 8;11(6):921. doi: 10.3390/cells11060921) recently published “Current Methods to Unravel the Functional Properties of Lysosomal Ion Channels and Transporters.”; this article should be mentioned in the introduction as a general reference on the methods to investigate the functions of lysosomal channels.

- lines 38-44: TPC2 is also involved in neoangiogenesis see Favia et al. (Proc Natl Acad Sci U S A. 2014 111(44):E4706-15. doi: 10.1073/pnas.1406029111). Please add this reference.

- “such as receptors (4) and viruses (5),”

as hypothesized first by Filippini et al. (Frontiers in Microbiology 2020 Apr 30;11:970. doi: 10.3389/fmicb.2020.00970) and then by Grimm and Tang (Cell Calcium. 2020 Jun;88:102212. doi: 10.1016/j.ceca.2020.102212), recent results indicate that TPC2 is involved in coronavirus infection, see Clementi et al. Pharmacological Research (2020) 105255; doi: 10.1016/j.phrs.2020.105255. These three references should be added in the manuscript.

- line 95 “e.g., vauolcin-1,”

please correct.

- line 162-163 “Increases in luminal pH stimulated voltage-activation of TPC1 (54) but see (49) (section 5).”

This sentence is not complete.

- lines 296-297 “Structural analyses of humanized TPC1 rationalized Na+ selectivity at the atomic level”

change to

Structural analyses of humanized AtTPC1 rationalized Na+ selectivity at the atomic level.

- Evaluation of human TPC1 single channel conductance by non stationary noise analysis was performed by Milenkovic et al. (Phys Chem Chem Phys. 2020 22(27):15664-15674. doi: 10.1039/d0cp00805b), in Fig. 9, see also Fig. S8 in electronic supplementary information. The single channel conductance was 5.5+/-1.6 pS, in agreement with the theoretical value estimated in that work. These measurements and reference should be added both in section 5 and table 2.

Author Response

Reviewer #1

In this review, Patel at al. focused on the electrophysiological recordings of two-pore channels TPC1 and TPC2, highlighting known and novel intriguing experimental results. The paper is clear, easy to follow and very informative.

We thank the reviewer for their comments.

There are some references, which should be included, and some minor changes to be made.

- Festa et al. (Cells. 2022 Mar 8;11(6):921. doi: 10.3390/cells11060921) recently published “Current Methods to Unravel the Functional Properties of Lysosomal Ion Channels and Transporters.”; this article should be mentioned in the introduction as a general reference on the methods to investigate the functions of lysosomal channels.

 Done

- lines 38-44: TPC2 is also involved in neoangiogenesis see Favia et al. (Proc Natl Acad Sci U S A. 2014 111(44):E4706-15. doi: 10.1073/pnas.1406029111). Please add this reference.

 Done.

- “such as receptors (4) and viruses (5),”

as hypothesized first by Filippini et al. (Frontiers in Microbiology 2020 Apr 30;11:970. doi: 10.3389/fmicb.2020.00970) and then by Grimm and Tang (Cell Calcium. 2020 Jun;88:102212. doi: 10.1016/j.ceca.2020.102212), recent results indicate that TPC2 is involved in coronavirus infection, see Clementi et al. Pharmacological Research (2020) 105255; doi: 10.1016/j.phrs.2020.105255. These three references should be added in the manuscript.

It was not our aim to be comprehensive here but just to highlight the first study. The link between TPCs and coronaviruses was actually established by the Marchant lab for SARS-CoV-1 prior to the studies mentioned by the reviewer (XX). And substantiated with data related to SARS-CoV-2 (XX).

- line 95 “e.g., vauolcin-1,”

please correct.

 Done.

- line 162-163 “Increases in luminal pH stimulated voltage-activation of TPC1 (54) but see (49) (section 5).”

This has been corrected.- lines 296-297 “Structural analyses of humanized TPC1 rationalized Na+ selectivity at the atomic level”

change to

Structural analyses of humanized AtTPC1 rationalized Na+ selectivity at the atomic level.

Done.

- Evaluation of human TPC1 single channel conductance by non stationary noise analysis was performed by Milenkovic et al. (Phys Chem Chem Phys. 2020 22(27):15664-15674. doi: 10.1039/d0cp00805b), in Fig. 9, see also Fig. S8 in electronic supplementary information. The single channel conductance was 5.5+/-1.6 pS, in agreement with the theoretical value estimated in that work. These measurements and reference should be added both in section 5 and table 2.

 Done.

Reviewer 2 Report

The authors describe the current knowledge on TPCs obtained via various electrophysiology approaches. The review is timely and of broad interest. Before publication the authors should still address a few comments below:

The authors describe very briefly the overall structure of TPCs. Could you please describe in more detail as it’s relevant in the following sections. Moreover, a scheme showing the overall structure together with critical modulatory regions and factor would be helpful for the reader. Even scheme showing the potential signaling pathways would increase the impact of the review.

Could you please provide a scheme/flow diagram for endolysosome patch-clamp methods.

Figure 1: The authors distinguish between macroscopic and microscopic electrophysiology methods. However, inside-out is ranked as macroscopic, however, I would suggest to put it to microscopic methods, as also stated in the text (2.2). Moreover, for planar patch-clamp, vacuolar patch-clamp and whole cell it could be a bit misleading that the reader thinks a single ion channel current is recorded. In this case I suggest to add a few more channels to highlight that currents are measured across all channels in the membrane.

Please provide a table for the different TPC variants and the modes of action of various reviewed factors: NAADP, PIP2, Mg2+, JNK

Do you have any hypothesis on controversial findings of human and plant TPCs eg. the role of Ca2+ (l. 180-187), different single channel conductances, different response to activators.

Is there any knowledge on potential binding pockets of the pharmacological compounds available?

Could the authors highlight the advantages and disadvantages of conventional patch-clamp analysis and single channel analysis of TPCs?

l. 127    single channel

l. 223 What are GWAS studies? Please describe?

l. 230 What does SNP mean?

l. 491 TPC3 …. Is this a mistake?

Throughout the review the authors put the reference at the end of the sentence, though stating in the text for instance Jha et al.. In these cases I suggest to put the reference immediately after xx et al. (Ref) instead of putting the Ref# at the end of the sentence.

Author Response

Reviewer #2

The authors describe the current knowledge on TPCs obtained via various electrophysiology approaches. The review is timely and of broad interest. Before publication the authors should still address a few comments below:

We thank the reviewer for their comments.

The authors describe very briefly the overall structure of TPCs. Could you please describe in more detail as it’s relevant in the following sections. Moreover, a scheme showing the overall structure together with critical modulatory regions and factor would be helpful for the reader. Even scheme showing the potential signaling pathways would increase the impact of the review.

We have now added a new Figure (Fig. 1) which is described in the Introduction depicting the structure and activation of TPC2.

Could you please provide a scheme/flow diagram for endolysosome patch-clamp methods.

This has been described in detail in other referenced work including an article in this issue (see comments to Reviewer #1).

Figure 1: The authors distinguish between macroscopic and microscopic electrophysiology methods. However, inside-out is ranked as macroscopic, however, I would suggest to put it to microscopic methods, as also stated in the text (2.2). Moreover, for planar patch-clamp, vacuolar patch-clamp and whole cell it could be a bit misleading that the reader thinks a single ion channel current is recorded. In this case I suggest to add a few more channels to highlight that currents are measured across all channels in the membrane.

The inside-out configuration features in both the macroscopic and microscopic sections of Figure 1. This is because it has been used for macropatch (see section 2.1 and 4) as well as single channel recordings. The figure has been updated to lessen ambiguity relating to single v multi-channel recordings.

Please provide a table for the different TPC variants and the modes of action of various reviewed factors: NAADP, PIP2, Mg2+, JNK

We do not think there is enough information yet across TPC isoforms  to summarise the regulatory factors, although we did summarise single channel conductance (Table 1) and ion selectivity (Table 3).

Do you have any hypothesis on controversial findings of human and plant TPCs eg. the role of Ca2+ (l. 180-187), different single channel conductances, different response to activators.

No!

Is there any knowledge on potential binding pockets of the pharmacological compounds available?

We have added a description in section 6 describing the requirement of K204 for the action of some TPC2 agonists. Otherwise we know nothing.

Could the authors highlight the advantages and disadvantages of conventional patch-clamp analysis and single channel analysis of TPCs?

We feel this is beyond the scope of this review.

  1. 127    single channel

Corrected.

  1. 223 What are GWAS studies? Please describe?

This abbreviation has been removed.

  1. 230 What does SNP mean?

This abbreviation has been removed.

  1. 491 TPC3 …. Is this a mistake?

This is correct. Please refer to references 83-86 (original draft numbering) for the identification and characterisation of TPC3.

Throughout the review the authors put the reference at the end of the sentence, though stating in the text for instance Jha et al.. In these cases I suggest to put the reference immediately after xx et al. (Ref) instead of putting the Ref# at the end of the sentence.

This will likely breach journal format.